# REFERENCE-BASED VARIATIONAL AUTOENCODERS

**Adria Ruiz**[*]    **Oriol Martinez**[†]    **Xavier Binefa**[†]    **Jakob Verbeek**[*]

[*] INIRA, Grenoble Rhone-Alpes, {adria.ruiz-ovejero,jakob.verbeek}@inria.fr
[†] Universitat Pompeu Fabra, Barcelona, {oriol.martinez,xavier.binefa}@upf.edu

## ABSTRACT

Learning disentangled representations from visual data, where different high-level generative factors are independently encoded, is of importance for many computer vision tasks. Solving this problem, however, typically requires to explicitly label all the factors of interest in training images. To alleviate the annotation cost, we introduce a learning setting which we refer to as *reference-based disentangling*. Given a pool of unlabelled images, the goal is to learn a representation where a set of target factors are disentangled from others. The only supervision comes from an auxiliary *reference set* containing images where the factors of interest are constant. To address this problem, we propose reference-based variational autoencoders, a novel deep generative model designed to exploit the weak-supervision provided by the reference set. By addressing tasks such as feature learning, conditional image generation or attribute transfer, we validate the ability of the proposed model to learn disentangled representations from this minimal form of supervision.

## 1 INTRODUCTION

Natural images are the result of a generative process involving a large number factors of variation. For instance, the appearance of a face is determined by the interaction between many latent variables including the subject's pose, illumination, identity, or expression. In this context, learning disentangled representations, where different generative factors are independently encoded in feature vectors, can be considered one of the most relevant problems in computer vision (Bengio et al., 2013). Recently, Variational auto-encoders (VAEs) (Kingma & Welling, 2014) have emerged as a powerful deep latent variable model able to address this task (Higgins et al., 2017; Kumar et al., 2018; Chen et al., 2018). However, VAEs are typically trained in an unsupervised manner and, therefore, they lack a mechanism to impose specific high-level semantics on the latent space. In order to address this limitation, different semi-supervised extensions have been proposed (Kingma et al., 2014; Narayanaswamy et al., 2017). However, these approaches require latent factors to be explicitly annotated in a training set in order to disentangle them.

In order to reduce the need of labelled generative factors, we introduce *reference-based disentangling*, a weakly-supervised learning setting in which, given a training set of unlabelled images, the goal is to learn a representation where a specific set of generative factors are disentangled from the rest. For that purpose, the only supervision comes in the form of an auxiliary *reference set* containing images where the factors of interest are constant (see Fig. 1a-b for illustrative examples). Note that a collection of reference images is generally easier to obtain compared to explicit labels of target factors. In order to address this problem, we present reference-based variational autoencoders (Rb-VAEs), a deep probabilistic model able to impose high-level semantics into the latent variables by only exploiting the weak-supervision provided by the reference set. An extended version of this work can be found at: https://arxiv.org/abs/1901.08534.

## 2 REFERENCE-BASED VARIATIONAL AUTOENCODERS

**Problem formalization:** Consider a training set of unlabelled images (*e.g.* human faces) $\mathbf{x} \in \mathbb{R}^{W \times H \times 3}$ sampled from a given distribution $p^u(\mathbf{x})$. Our goal is to learn a latent variable model defining a joint distribution over $\mathbf{x}$ and latent variables $\mathbf{e} \in \mathbb{R}^{D_e}$ and $\mathbf{z} \in \mathbb{R}^{D_z}$. Whereas $\mathbf{e}$ is expected to encode information about a set of generative factors of interest, *e.g.* facial expressions, $\mathbf{z}$ should model the remaining factors of variation underlying the images, *e.g.* pose, illumination, identity, *etc*. From now on, we will refer to $\mathbf{e}$ and $\mathbf{z}$ as the "target" and "common factors", respectively. In order to disentangle them, we are provided with an additional set of reference images sampled from

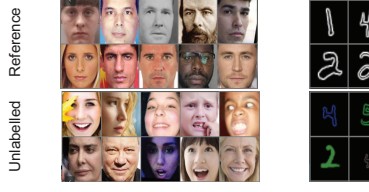 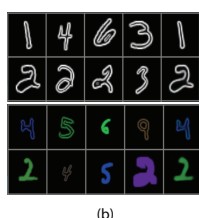 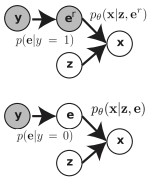 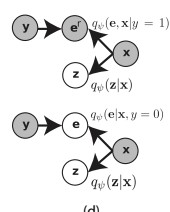

Figure 1: (a) Disentangling factors underlying facial expression. The reference set contains faces with neutral expression. (b) Disentangling style from digits. The reference set is composed by digits with a fixed style.(c) Rb-VAE generative process where $p_\theta(\mathbf{x}|\mathbf{z}, \mathbf{e})$ maps $\mathbf{z}$ (common factors) and $\mathbf{e}$ (target factors) to images $\mathbf{x}$. (d) Approximate posteriors $q(\mathbf{z}|x)$ and $q(\mathbf{e}|\mathbf{x}, y)$ map images $\mathbf{x}$ to the corresponding common and target factors $\mathbf{z}$ and $\mathbf{e}$ respectively.

$p^r(\mathbf{x})$, representing a distribution over $\mathbf{x}$ where target factors $\mathbf{e}$ are constant *e.g.* neutral faces. Given $p^r(\mathbf{x})$ and $p^u(\mathbf{x})$, we define a binary variable $y$ indicating whether an image $\mathbf{x}$ has been sampled from the unlabelled or reference distributions, *i.e.* $p(\mathbf{x}|y = 0) = p^u(\mathbf{x})$ and $p(\mathbf{x}|y = 1) = p^r(\mathbf{x})$.

**Model definition:** Rb-VAEs is a deep latent variable model defining a joint distribution $p_\theta(\mathbf{x}, \mathbf{z}, \mathbf{e}, y) = p_\theta(\mathbf{x}|\mathbf{z}, \mathbf{e})p(\mathbf{z})p(\mathbf{e}|y)p(y)$, where conditional dependencies are designed to address the reference-based disentangling problem (see Fig. 1c). In particular, we define $p_\theta(\mathbf{x}|\mathbf{z}, \mathbf{e}) = \mathcal{L}(\mathbf{x}|\mathcal{G}_\theta(\mathbf{z}, \mathbf{e}), \lambda)$, where $\mathcal{G}_\theta(\mathbf{z}, \mathbf{e})$ is a generator network, mapping a pair of latent variables $(\mathbf{z}, \mathbf{e})$ to an image defining the mean of a Laplace distribution $\mathcal{L}$ with fixed scale parameter $\lambda$. To reflect the assumption of constant target factors across reference images, we define the conditional distribution over $\mathbf{e}$ given $y = 1$ as a delta peak centered on a learned vector $\mathbf{e}^r \in R^{D_e}$, *i.e.* $p(\mathbf{e}|y = 1) = \delta(\mathbf{e} - \mathbf{e}^r)$. In contrast, for unlabelled images, $p(\mathbf{e}|y = 0) = \mathcal{N}(\mathbf{e}|\mathbf{0}, \mathbf{I})$ as in standard VAEs. In the following, we denote $p(\mathbf{e}|y = 0) = p(\mathbf{e})$. Contrary to the case of target factors $\mathbf{e}$, the prior $p(\mathbf{z})$ over common factors is equal for reference and unlabelled images, and taken to be a unit Gaussian. Finally, we assume $p(y = 0) = p(y = 1) = \frac{1}{2}$.

**Conventional Variational Learning:** We define a variational distribution $q_\psi(\mathbf{x}, \mathbf{z}, \mathbf{e}, y) = q_\psi(\mathbf{x}, \mathbf{z}|\mathbf{x})q_\psi(\mathbf{e}|\mathbf{x}, y)p(\mathbf{x}, y)$, where the approximate posteriors $q_\psi(\mathbf{e}|\mathbf{x}, y)$ and $q_\psi(\mathbf{z}|\mathbf{x})$ allow to infer target and common factors $\mathbf{e}$ and $\mathbf{z}$ given an image $\mathbf{x}$ (see Fig. 1d). For reference images, *i.e.* with $y = 1$, the target factors $q_\psi(\mathbf{e}|\mathbf{x}, y = 1)$ are known to be equal to the reference value $\mathbf{e}^r$. On the other hand, given an unlabelled image, *i.e.* with $y = 0$, we define the approximate posterior $q_\psi(\mathbf{e}|\mathbf{x}, y = 0) = \mathcal{N}(\mathbf{e}|\mathcal{E}^\mu(\mathbf{x}), \mathcal{E}^\sigma(\mathbf{x}))$, where the mean and diagonal covariance matrices of a conditional Gaussian distribution are given by the output of deep neural network $\mathcal{E}$. Similarly, we use an additional network $\mathcal{Z}$ to model $q_\psi(\mathbf{z}|\mathbf{x}) = \mathcal{N}(\mathbf{z}|\mathcal{Z}^\mu(\mathbf{x}), \mathcal{Z}^\sigma(\mathbf{x}))$. Following the standard training strategy employed in VAEs, we learn parameters $\theta$ and $\psi$ by minimizing the KL divergence between $p_\theta$ and $q_\psi$, which, as shown in Appendix A.1, is equivalent to:

$$\min_{\theta, \psi, \mathbf{e}^r} \quad \mathbb{E}_{p^u(\mathbf{x})} \Big[ \mathbb{KL}(q_\psi(\mathbf{z}|\mathbf{x})q_\psi(\mathbf{e}|\mathbf{x}) \parallel p(\mathbf{z})p(\mathbf{e})) - \mathbb{E}_{q_\psi(\mathbf{z}|\mathbf{x})q_\psi(\mathbf{e}|\mathbf{x})} \log(p_\theta(\mathbf{x}|\mathbf{z}, \mathbf{e})) \Big] +$$

$$\mathbb{E}_{p^r(\mathbf{x})} \Big[ \mathbb{KL}(q_\psi(\mathbf{z}|\mathbf{x}) \parallel p(\mathbf{z})) - \mathbb{E}_{q_\psi(\mathbf{z}|\mathbf{x})} \log(p_\theta(\mathbf{x}|\mathbf{z}, \mathbf{e}^r)) \Big], \tag{1}$$

The second and fourth terms correspond to the reconstruction errors for unlabelled and reference images respectively. Note that, for reference images, no inference over target factors $\mathbf{e}$ is needed. Instead, the generator reconstructs them using the learned parameter $\mathbf{e}^r$. The remaining terms consist of KL divergences between approximate posteriors and priors over latent variables. The minimization in Eq. (1) can be solved using SGD and the *re-parametrization* trick (Rezende et al., 2014), in order to back-propagate the gradient when sampling from $q_\psi(\mathbf{e}|\mathbf{x})$ and $q_\psi(\mathbf{z}|\mathbf{x})$.

**Symmetric Variational Learning**: The main limitation of the conventional variational learning for Rb-VAEs is that it does not guarantee that common and target factors will be effectively disentangled in $\mathbf{z}$ and $\mathbf{e}$ respectively. In particular, the minimization of Eq. (1) does not prevent the degenerate solution $p_\theta(\mathbf{x}|\mathbf{z}, \mathbf{e}) = p_\theta(\mathbf{x}|\mathbf{z})$, where the inferred latent variables by $q_\psi(\mathbf{e}|\mathbf{x})$ are ignored and target and common factors are jointly encoded into $\mathbf{z}$. To address this limitation, we optimize an alternative variational expression inspired by Symmetric VAEs (Pu et al., 2018). Concretely, we add the reversed KL divergence between $p_\theta$ and $q_\psi$ to the minimized objective. In order to understand why this additional term allows to mitigate the degenerate solution $p_\theta(\mathbf{x}|\mathbf{z}, \mathbf{e}) = p_\theta(\mathbf{x}|\mathbf{z})$, it is necessary to observe that its minimization is equivalent to:

| | AffectNet | | | | | | | | MNIST | | | | | |
|---|---|---|---|---|---|---|---|---|---|---|---|---|---|---|
| | **Happ** | **Sad** | **Sur** | **Fear** | **Disg** | **Ang** | **Compt** | **Avg.** | **R** | **G** | **B** | **Scale** | **Width** | **Avg.** |
| VAE | .554 | .279 | .383 | .357 | .256 | .415 | .439 | .383 | .099 | .104 | .101 | [.034] | **.085** | .085 |
| DIP-VAE-I | .561 | .269 | [.401] | **.367** | .258 | .397 | .463 | **.388** | [.055] | **.064** | .063 | .038 | .100 | **.064** |
| DIP-VAE-II | .548 | .245 | [.401] | [.389] | .268 | .391 | .463 | .386 | .077 | .069 | .076 | **.035** | .098 | .071 |
| $\beta$ VAE | .581 | .283 | .373 | .323 | .250 | .415 | .467 | .384 | .093 | .099 | .094 | .039 | .089 | .083 |
| sVAE | **.583** | .251 | .389 | .349 | .260 | .391 | .469 | .384 | .094 | .092 | .084 | .036 | .104 | .082 |
| $\beta$-TCVAE | .563 | .277 | **.393** | .349 | .256 | [.427] | .467 | .390 | .098 | .100 | .099 | [.034] | [.084] | .083 |
| [Mathieu et. al] | .567 | .388 | .312 | .330 | .295 | .353 | [.512] | **.395** | .116 | .116 | .114 | .039 | .104 | .098 |
| Rb-VAE | .536 | **.393** | .379 | .311 | **.320** | .383 | .421 | 392 | .065 | .069 | **.062** | .061 | .095 | .070 |
| sRb-VAE | [.587] | [.405] | .387 | .327 | [.344] | **.425** | **.483** | [.422] | .057 | [.053] | [.055] | .038 | .095 | [.060] |

Table 1: Prediction of target factors from learned representations. We report accuracy (AffectNet) and mean-absolute-error (MNIST). Two best methods shown in bold, best result in brackets.

$$\min_{\theta,\psi} \ \mathbb{E}_{p(\mathbf{z},\mathbf{e})}\Big[\mathbb{KL}(p_\theta(\mathbf{x}|\mathbf{z},\mathbf{e}) \parallel p^u(\mathbf{x})) - \mathbb{E}_{p_\theta(\mathbf{x}|\mathbf{z},\mathbf{e})}[\log(q_\psi(\mathbf{z}|\mathbf{x})) + \log(q_\psi(\mathbf{e}|\mathbf{x}))]\Big] +$$
$$\mathbb{E}_{p(\mathbf{z})p_\theta(\mathbf{x}|\mathbf{z},\mathbf{e}^r)}\Big[\mathbb{KL}(p_\theta(\mathbf{x}|\mathbf{z},\mathbf{e}^r)||p^r(\mathbf{x})) - \log(q_\psi(\mathbf{z}|\mathbf{x}))\Big], \tag{2}$$

see Appendix A.1 for details. Note that the second and fourth terms correspond to reconstruction errors over latent variables $\mathbf{z}, \mathbf{e}$ inferred from generated images drawn from $p_\theta$. As a consequence, the minimization of these errors is encouraging the generator $p_\theta(\mathbf{x}|\mathbf{z},\mathbf{e})$ to generate images $\mathbf{x}$ by taking into account latent variables $\mathbf{e}$, since the latter must be reconstructed via $q_\psi(\mathbf{e}|\mathbf{x})$. Therefore, the minimization of the reversed KL avoids the degenerate solution ignoring $\mathbf{e}$.

In order to jointly optimize Eqs. (1) and (2), we propose to use an adversarial learning procedure. For this purpose, we express the minimization of the two KL divergences as:

$$\min_{\theta,\psi} \ \mathbb{E}_{q_\psi(\mathbf{e},\mathbf{z}|\mathbf{x})p^u(\mathbf{x})}\mathcal{L}_{\mathbf{xze}} - \mathbb{E}_{p_\theta(\mathbf{x}|\mathbf{e},\mathbf{z})p(\mathbf{z})p(\mathbf{e})}\mathcal{L}_{\mathbf{xze}} + \mathbb{E}_{q_\psi(\mathbf{z}|\mathbf{x})p^r(\mathbf{x})}\mathcal{L}_{\mathbf{xz}} - \mathbb{E}_{p_\theta(\mathbf{x}|\mathbf{e}^r,\mathbf{z})p(\mathbf{z})}\mathcal{L}_{\mathbf{xz}},$$

where $\mathcal{L}_{\mathbf{xze}}$ corresponds to the log-density ratio between distributions $q_\psi(\mathbf{e},\mathbf{z}|\mathbf{x})p^u(\mathbf{x})$ and $p_\theta(\mathbf{x}|\mathbf{e},\mathbf{z})p(\mathbf{z})p(\mathbf{e})$. Similarly, $\mathcal{L}_{\mathbf{xz}}$ defines an analogous expression for $q_\psi(\mathbf{z}|\mathbf{x})p^r(\mathbf{x})$ and $p_\theta(\mathbf{x}|\mathbf{e}^r,\mathbf{z})p(\mathbf{z})$ (see Appendix A.1 for details). The defined expression can be minimized by evaluating ($\mathcal{L}_{\mathbf{xze}}$, $\mathcal{L}_{\mathbf{xz}}$) and back-propagating the gradients w.r.t. parameters $\psi$ and $\theta$ using the reparametrization-trick over samples $(\mathbf{x}, \mathbf{z}, \mathbf{e})$. During this procedure, two auxiliary parametric functions $d_\xi(\mathbf{x}, \mathbf{z}, \mathbf{e})$ and $d_\gamma(\mathbf{x}, \mathbf{z})$ are used to approximate $\mathcal{L}_{\mathbf{xze}}$ and $\mathcal{L}_{\mathbf{xz}}$, respectively. These functions are implemented as deep convolutional networks and are analogous to the discriminators used in adversarial methods such as ALICE (Li et al., 2017), where the function $d_\gamma(\cdot)$ is trained as a classifier trying to distinguish whether pairs of reference images $\mathbf{x}$ and latent variables $\mathbf{z}$ have been generated by $q_\psi$ or $p_\theta$. However, in our case we have an additional discriminator $d_\xi$ operating over unlabelled images and its corresponding latent variables $\mathbf{z}$ and $\mathbf{e}$. Similar to Symmetric VAEs (Pu et al., 2018), we also add into the minimized objective the reconstruction terms in Eqs. (1) and (2) for images and inferred latent variables. See Appendix A.2 for an illustration of the learning algorithm.

## 3 EXPERIMENTS

**Datasets:** In our experiments, we consider two different reference-based disentangling tasks. In the first case, the goal is to model style variations (scale, width and color) from hand-written digits. We use half of the original training images in the MNIST dataset (LeCun et al., 1998) as our reference distribution (30k examples). The unlabelled set is synthetically generated by applying two different transformations over the remaining half of images simulating random digit styles (60k images). In the second problem, we address the disentangling of facial expressions by using a reference set of neutral faces. As unlabelled images, we use a subset (150k samples) of the AffectNet dataset (Mollahosseini et al., 2017) . Some of these images are annotated according to different facial expressions: *happiness, sadness, surprise, fear, disgust, anger*, and *contempt*. As our reference set, we collected a set of neutral faces (10k samples). See more details in Appendix A.3.

**Baselines:** We evaluate the two different variants of our proposed method: Rb-VAE, trained using the standard variational objective, and sRb-VAE, learned by minimizing the symmetric KL divergence. We compare both methods with state-of-the-art unsupervised approaches based on the VAE framework: $\beta$-VAE (Higgins et al., 2017), $\beta$-TCVAE (Chen et al., 2018), sVAE (Pu et al., 2018), DIP-VAE (Kumar et al., 2018) and vanilla VAEs (Kingma & Welling, 2014). Finally, in order to evaluate an alternative weakly-supervised baseline exploiting the reference-set, we have implemented (Mathieu et al., 2016) adapting it to our context. For a fair comparison, we have developed

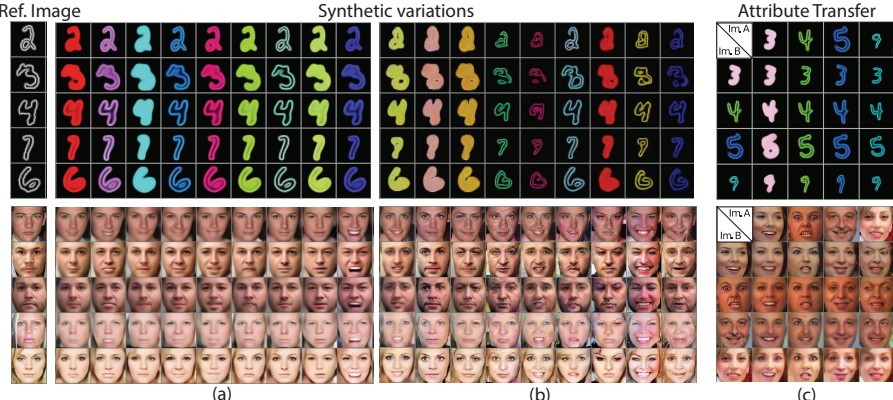

Figure 2: Conditional image synthesis using Rb-VAE (a) and sRb-VAE (b). Within each column images are generated using the same random target factors **e**. (c) Transferring target factors **e** from image A to an image B with sRb-VAE.

our own implementation for all the evaluated methods in order to use the same network architectures and hyper-parameters. Concretely, for encoders, generators and discriminators we use standard conv-deconv architectures employing the main building blocks used by Karras et al. (2018). See Appendix A.4 for more details about the specific architectures.

**Feature Learning:** Following a similar evaluation than Mathieu et al. (2016), we use the learned representations as feature vectors to estimate the target factors involved in each problem. In MNIST we employ a set of linear-regressors predicting the scale, width and color parameters for each digit. To predict the different expression classes in the AffectNet dataset, we use a linear classifier. For methods using the reference-set, we used only the inferred latent variables **e** as features given that they are expected to encode the information regarding the target factors. For evaluation, we split each dataset in three subsets. The first is used to learn each generative model. Then, the second is used for training the regressors or classifier. Finally, the third is used to evaluate the predictions in terms of the mean absolute error and per-class accuracy for the MNIST and AffectNet, respectively. Table 1 shows the results obtained by the different evaluated methods. Note that the unsupervised approach DIP-VAE-I achieves better average results than Rb-VAE for MNIST. Moreover, in Affect-Net, $\beta$-TCVAE achieves comparable or better performance in several cases. This may seem counter-intuitive because, unlike Rb-VAE, DIP-VAE-I is trained without the weak-supervision provided by reference images. However, it confirms that the learning objective of Rb-VAE does not explicitly encourage the disentanglement between target and common factors. In contrast, we can see that in most cases sRb-VAE obtains comparable or better results than rest of the methods. Moreover, it achieves the best average performance in both datasets. This demonstrates that the information provided by the reference distribution is effectively exploited by the symmetric KL objective used to train sRb-VAE. To conclude, note that sRb-VAE also obtains better performance than Mathieu et al. (2016) in both data-sets. So even though this method also uses reference-images during training, sRb-VAE is shown to better exploit the weak-supervision in reference-based disentangling.

**Conditional image generation:** In this task, the goal is to transform real images by modifying only the target factors. For this purpose, we use the generator network to obtain a new image from two vectors: **z**, inferred from the original image, and **e**, randomly sampled from $\mathcal{N}(\mathbf{0}, \mathbf{1})$. Fig. 2a-b shows images generated by sRb-VAE and Rb-VAE. As we can observe, sRb-VAE generates more convincing results than its non-symmetric counterpart. In AffectNet, the amount of variability in Rb-VAE samples is quite low. In contrast, sRb-VAE is able to generate more diverse expressions. Similarly, Rb-VAE is not able to model scale variations in the MNIST database, while sRb-VAE does. Again, this shows the benefits of using the proposed symmetric objective.

**Attribute transfer:** In this problem, we aim to swap the target factors **e** between a pair of images A and B. Fig. 2c shows images generated by sRb-VAE in this case. Note that our model is able to effectively transfer only the expression and the digit style in the AffectNet and MNIST datasets respectively, while keeping the rest of generative factors unchanged. Again, this shows that our method is able to disentangle common and target factors by using only the weak-supervision provided by the reference-set.

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

## Appendix A    Appendix

### A.1    Mathematical derivations

**Equivalence between** $\mathbb{KL}(q_\psi(\mathbf{x}, \mathbf{z}, \mathbf{e}, y) \parallel p_\theta(\mathbf{x}, \mathbf{z}, \mathbf{e}, y))$ **and Eq. (1)**

$$\sum_{y \in [0,1]} \int_{x,e,z} q_\psi(\mathbf{e}|\mathbf{x}, y) q_\psi(\mathbf{z}|\mathbf{x}) p(\mathbf{x}|y) p(y) \log \left( \frac{q_\psi(\mathbf{z}, \mathbf{e}|\mathbf{x}, y) p(\mathbf{x}|y) p(y)}{p_\theta(\mathbf{x}|\mathbf{e}, \mathbf{z}) p(\mathbf{z}) p(\mathbf{e}|y) p(y)} \right) d\mathbf{x} d\mathbf{z} d\mathbf{e} \quad (3)$$

$$= \frac{1}{2} \int_{x,e,z} q_\psi(\mathbf{e}|\mathbf{x}) q_\psi(\mathbf{z}|\mathbf{x}) p^u(\mathbf{x}) \log \left( \frac{q_\psi(\mathbf{e}|\mathbf{x}) q_\psi(\mathbf{z}|\mathbf{x}) p^u(\mathbf{x})}{p_\theta(\mathbf{x}|\mathbf{e}, \mathbf{z}) p(\mathbf{z}) p(\mathbf{e})} \right) d\mathbf{x} d\mathbf{z} d\mathbf{e}$$

$$+ \frac{1}{2} \int_{x,z} q_\psi(\mathbf{z}|\mathbf{x}) p^r(\mathbf{x}) \log \left( \frac{q_\psi(\mathbf{z}|\mathbf{x}) p^r(\mathbf{x})}{p_\theta(\mathbf{x}|\mathbf{e}^r, \mathbf{z}) p(\mathbf{z})} \right) d\mathbf{x} d\mathbf{z} \quad (4)$$

$$= \frac{1}{2} \mathbb{E}_{p^u(\mathbf{x})} \mathbb{E}_{q_\psi(\mathbf{e}|\mathbf{x}) q_\psi(\mathbf{z}|\mathbf{x})} \left[ \log \left( \frac{q_\psi(\mathbf{e}|\mathbf{x}) q_\psi(\mathbf{z}|\mathbf{x})}{p(\mathbf{z}) p(\mathbf{e})} \right) - \log(p_\theta(\mathbf{x}|\mathbf{e}, \mathbf{z})) \right] - H^u(\mathbf{x})$$

$$+ \frac{1}{2} \mathbb{E}_{p^r(\mathbf{x})} \mathbb{E}_{q_\psi(\mathbf{z}|\mathbf{x})} \left[ \log \left( \frac{q_\psi(\mathbf{z}|\mathbf{x})}{p(\mathbf{z})} \right) - \log(p_\theta(\mathbf{x}|\mathbf{e}^r, \mathbf{z})) \right] - H^r(\mathbf{x}) \quad (5)$$

$$= \frac{1}{2} \mathbb{E}_{p^u(\mathbf{x})} \left[ \mathbb{KL}(q_\psi(\mathbf{z}|\mathbf{x}) q_\psi(\mathbf{e}|\mathbf{x}) \parallel p(\mathbf{z}) p(\mathbf{e})) - \mathbb{E}_{q_\psi(\mathbf{z}|\mathbf{x}) q_\psi(\mathbf{e}|\mathbf{x})} \log(p_\theta(\mathbf{x}|\mathbf{z}, \mathbf{e})) \right]$$

$$+ \frac{1}{2} \mathbb{E}_{p^r(\mathbf{x})} \left[ \mathbb{KL}(q_\psi(\mathbf{z}|\mathbf{x}) \parallel p(\mathbf{z})) - \mathbb{E}_{q_\psi(\mathbf{z}|\mathbf{x})} \log(p_\theta(\mathbf{x}|\mathbf{z}, \mathbf{e}^r)) \right] - H^r(\mathbf{x}) - H^u(\mathbf{x}) \quad (6)$$

We use $H^r(X)$ and $H^u(X)$ to denote the entropy of the reference and unlabelled distributions $p^r(\mathbf{x})$ and $p^u(\mathbf{x})$ respectively. Note that they can be ignored during the minimization since are constant w.r.t. parameters $\theta$ and $\psi$. For the second equality, we have used the definitions $p(\mathbf{x}|y = 0) = p^u(\mathbf{x})$, $p(\mathbf{x}|y = 1) = p^r(\mathbf{x})$ and assumed $p(y = 0) = p(y = 1) = \frac{1}{2}$. Moreover, we have exploited the fact that $q_\psi(\mathbf{e}|\mathbf{x}, y = 1)$ and $p(\mathbf{e}|y = 1)$ are defined as delta functions and, therefore, $\mathbb{E}_{p(\mathbf{e}|y=1)} log(\frac{p(\mathbf{e}|y=1)}{q_\psi(\mathbf{e}|y=1)}) = 0$. We denote $p(\mathbf{e}|y = 0) = p(\mathbf{e})$ and $q_\psi(\mathbf{e}|\mathbf{x}, y = 0) = q_\psi(\mathbf{e}|\mathbf{x})$ for the sake of brevity.

**Equivalence between** $\mathbb{KL}(p_\theta(\mathbf{x}, \mathbf{z}, \mathbf{e}, y) \parallel q_\psi(\mathbf{x}, \mathbf{z}, \mathbf{e}, y))$ **and the expression in Eq. (2)**

$$\sum_{y \in [0,1]} \int_{x,e,z} p_\theta(\mathbf{x}|\mathbf{e}, \mathbf{z}) p(\mathbf{z}) p(\mathbf{e}|y) p(y) \log \left( \frac{p_\theta(\mathbf{x}|\mathbf{e}, \mathbf{z}) p(\mathbf{z}) p(\mathbf{e}|y) p(y)}{q_\psi(\mathbf{z}, \mathbf{e}|\mathbf{x}, y) p(\mathbf{x}|y) p(y)} \right) d\mathbf{x} d\mathbf{z} d\mathbf{e} \quad (6)$$

$$= \frac{1}{2} \int_{x,e,z} p_\theta(\mathbf{x}|\mathbf{e}, \mathbf{z}) p(\mathbf{z}) p(\mathbf{e}) \log \left( \frac{p_\theta(\mathbf{x}|\mathbf{e}, \mathbf{z}) p(\mathbf{z}) p(\mathbf{e})}{q_\psi(\mathbf{e}|\mathbf{x}) q_\psi(\mathbf{z}|\mathbf{x}) p^u(\mathbf{x})} \right) d\mathbf{x} d\mathbf{z} d\mathbf{e}$$

$$+ \frac{1}{2} \int_{x,z} p_\theta(\mathbf{x}|\mathbf{e}^r, \mathbf{z}) p(\mathbf{z}) \log \left( \frac{p_\theta(\mathbf{x}|\mathbf{e}^r, \mathbf{z}) p(\mathbf{z})}{q_\psi(\mathbf{z}|\mathbf{x}) p^r(\mathbf{x})} \right) d\mathbf{x} d\mathbf{z} d\mathbf{e} \quad (7)$$

$$= \frac{1}{2} \mathbb{E}_{p(\mathbf{z}) p(\mathbf{e})} \mathbb{E}_{p_\theta(\mathbf{x}|\mathbf{e}, \mathbf{z})} \left[ \log \left( \frac{p_\theta(\mathbf{x}|\mathbf{e}, \mathbf{z})}{p(\mathbf{x})^u} \right) - \log(q_\psi(\mathbf{e}|\mathbf{x}) q_\psi(\mathbf{z}|\mathbf{x})) \right]$$

$$+ \frac{1}{2} \mathbb{E}_{p(\mathbf{z})} \mathbb{E}_{p_\theta(\mathbf{x}|\mathbf{e}^r, \mathbf{z})} \left[ \log \left( \frac{p_\theta(\mathbf{x}|\mathbf{e}^r, \mathbf{z})}{p(\mathbf{x})^r} \right) - \log(q_\psi(\mathbf{z}|\mathbf{x})) \right] - H(\mathbf{z}) - \frac{1}{2} H(\mathbf{e}) \quad (8)$$

$$= \frac{1}{2} \mathbb{E}_{p(\mathbf{z}) p(\mathbf{e})} \left[ \mathbb{KL}(p_\theta(\mathbf{x}|\mathbf{z}, \mathbf{e}) \parallel p^u(\mathbf{x})) - \mathbb{E}_{p_\theta(\mathbf{x}|\mathbf{z}, \mathbf{e})} [\log(q_\psi(\mathbf{z}|\mathbf{x})) + \log(q_\psi(\mathbf{e}|\mathbf{x}))] \right]$$

$$+ \frac{1}{2} \mathbb{E}_{p(\mathbf{z})} \left[ \mathbb{KL}(p_\theta(\mathbf{x}|\mathbf{z}, \mathbf{e}^r) \parallel p^r(\mathbf{x})) - \mathbb{E}_{p_\theta(\mathbf{x}|\mathbf{z}, \mathbf{e}^r)} \log(q_\psi(\mathbf{z}|\mathbf{x})) \right] - H(\mathbf{z}) - \frac{1}{2} H(\mathbf{e}) \quad (9)$$

We have used the same definitions and assumptions previously discussed. Moreover, we denote $H(\mathbf{z})$ and $H(\mathbf{e})$ as the entropy of the priors $p(\mathbf{z})$ and $p(\mathbf{e})$. Again, we can ignore these terms when we are optimizing w.r.t parameters $\psi$ and $\theta$.

**Equivalence between the minimization of the symmetric KL divergence and the expression in Eq. (3)**

$$\mathbb{KL}(q_\psi(\mathbf{z}, \mathbf{e}, \mathbf{x}, y) \parallel p_\theta(\mathbf{x}, \mathbf{z}, \mathbf{e}, y)) + \mathbb{KL}(p_\theta(\mathbf{x}, \mathbf{z}, \mathbf{e}, y) \parallel q_\psi(\mathbf{z}, \mathbf{e}, \mathbf{x}, y)) = \tag{10}$$

$$= \mathbb{E}_{q_\psi(\mathbf{e}|\mathbf{x}, y)q_\psi(\mathbf{z}|\mathbf{x})p(\mathbf{x}|y)p(y)} \log \left( \frac{q_\psi(\mathbf{e}|\mathbf{x}, y)q_\psi(\mathbf{z}|\mathbf{x})p(\mathbf{x}|y)p(y)}{p_\theta(\mathbf{x}|\mathbf{e}, \mathbf{z})p(\mathbf{z})p(\mathbf{e}|y)p(y)} \right)$$

$$+ \mathbb{E}_{p_\theta(\mathbf{x}|\mathbf{e}, \mathbf{z})p(\mathbf{z})p(\mathbf{e}|y)p(y)} \log \left( \frac{p_\theta(\mathbf{x}|\mathbf{e}, \mathbf{z})p(\mathbf{z})p(\mathbf{e}|y)p(y)}{q_\psi(\mathbf{e}|\mathbf{x}, y)q_\psi(\mathbf{z}|\mathbf{x})p(\mathbf{x}|y)p(y)} \right) \tag{11}$$

$$= \frac{1}{2} \left[ \mathbb{E}_{q_\psi(\mathbf{e}, \mathbf{z}|\mathbf{x})p^u(\mathbf{x})} \log \left( \frac{q_\psi(\mathbf{e}, \mathbf{z}|\mathbf{x})p^u(\mathbf{x})}{p_\theta(\mathbf{x}|\mathbf{e}, \mathbf{z})p(\mathbf{z})p(\mathbf{e})} \right) + \mathbb{E}_{q_\psi(\mathbf{z}|\mathbf{x})p^r(\mathbf{x})} \log \left( \frac{q_\psi(\mathbf{z}|\mathbf{x})p(\mathbf{x})^r}{p_\theta(\mathbf{x}|\mathbf{e}^r, \mathbf{z})p(\mathbf{z})} \right) \right.$$

$$\left. + \mathbb{E}_{p_\theta(\mathbf{x}|\mathbf{e}, \mathbf{z})p(\mathbf{z})p(\mathbf{e})} \log \left( \frac{p_\theta(\mathbf{x}|\mathbf{e}, \mathbf{z})p(\mathbf{z})p(\mathbf{e})}{q_\psi(\mathbf{e}, \mathbf{z}|\mathbf{x})p^u(\mathbf{x})} \right) + \mathbb{E}_{p_\theta(\mathbf{x}|\mathbf{e}^r, \mathbf{z})p(\mathbf{z})} \log \left( \frac{p_\theta(\mathbf{x}|\mathbf{e}^r, \mathbf{z})p(\mathbf{z}))}{q_\psi(\mathbf{z}|\mathbf{x})p^r(\mathbf{x})} \right) \right] \tag{12}$$

$$= \frac{1}{2} \left[ \mathbb{E}_{q_\psi(\mathbf{e}, \mathbf{z}|\mathbf{x})p^u(\mathbf{x})} \mathcal{L}_{\mathbf{xze}} + \mathbb{E}_{q_\psi(\mathbf{z}|\mathbf{x})p^r(\mathbf{x})} \mathcal{L}_{\mathbf{xz}} - \mathbb{E}_{p_\theta(\mathbf{x}|\mathbf{e}, \mathbf{z})p(\mathbf{z})p(\mathbf{e})} \mathcal{L}_{\mathbf{xze}} - \mathbb{E}_{p_\theta(\mathbf{x}|\mathbf{e}^r, \mathbf{z})p(\mathbf{z})} \mathcal{L}_{\mathbf{xz}} \right] \tag{13}$$

## A.2 Learning Rb-VAEs with the symmetric KL divergence

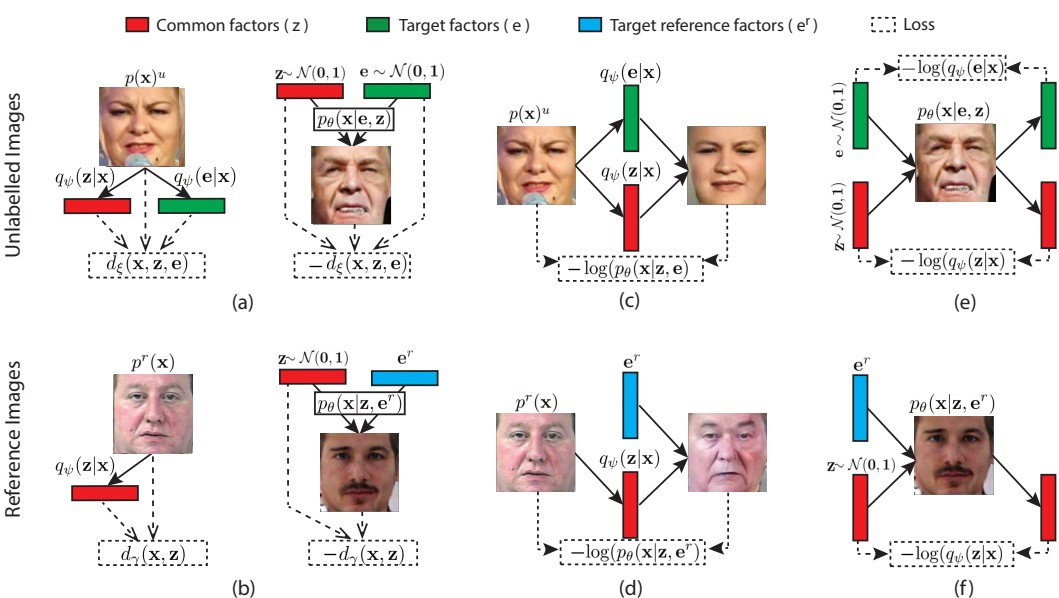

Figure 3: Losses used by sRB-VAE. Discriminator $d_\xi(\mathbf{x}, \mathbf{z}, \mathbf{e})$ measures the log-density ratio between the distributions $q_\psi(\mathbf{z}, \mathbf{e}|\mathbf{x})p^u(\mathbf{x})$ and $p_\theta(\mathbf{x}|\mathbf{e}, \mathbf{z})p(\mathbf{z})p(\mathbf{e})$. (b) Similar loss for reference images using an additional discriminator $d_\gamma(\mathbf{x}, \mathbf{z})$ (c,d) Reconstruction errors for unlabelled and reference images. (e,f) Reconstruction error over latent variables inferred from unlabelled and reference images generated using $p(\mathbf{z})$, $p(\mathbf{e})$ and $\mathbf{e^r}$

## A.3 Datasets

Examples of reference and unlabelled images for MNIST and AffectNet are shown in Fig. 4. In the following, we provide more information about the used datasets.

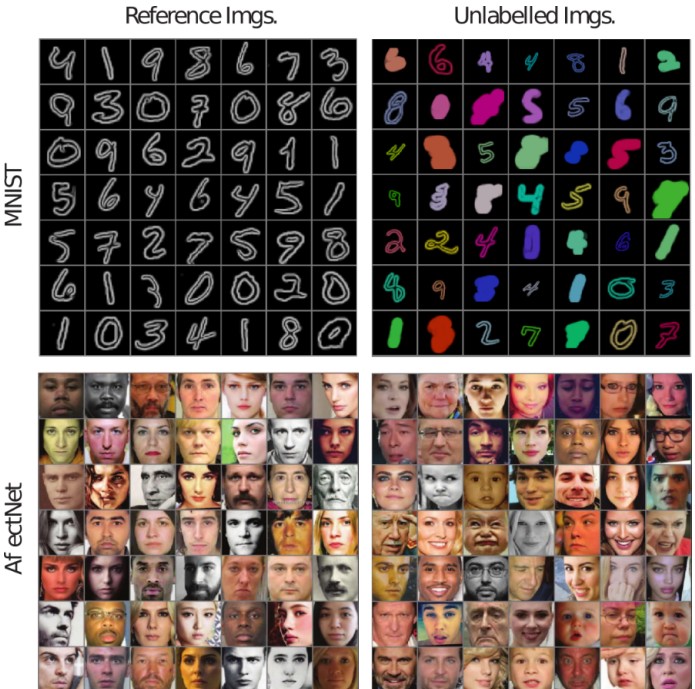

Figure 4: Examples of reference and unlabelled images used in our experiments. Extracted from MNIST (top) and AffectNet (bottom) datasets.

### A.3.1 MNIST

We use slightly modified version of the MNIST images: the size is increased to $64 \times 64$ pixels and an edge detection procedure is applied to keep only the boundaries of the digit. We obtain the samples in the unlabelled dataset by applying the following transformations over the MNIST images:

1. **Width**: Generate a random integer in the range $\{1, \ldots, 10\}$ using a uniform distribution. Apply a dilation operation over the image using a squared kernel with pixel-size equal to the generated number.

2. **Color**: Generate a random 3D vector $c \in [0, 1]^3$ using a uniform distribution. Normalize the resulting vector as $\hat{c} = c/||c||_1$. Multiply the RGB components of all the pixels in the image by $\hat{c}$.

3. **Size**: Generate a random number in the range $[0.5, 1]$ using a uniform distribution. Downscale the image by a factor equal to the generated number. Apply zero-padding to the resulting image in order to recover the original resolution.

### A.3.2 AFFECTNET

In order to remove 2D affine transformations such as scaling or in-plane rotations, we apply an alignment process to the face images. We localize facial landmarks using the approach of Xiong & De la Torre (2013). Then, we apply Procrustes analysis in order to find an affine transformation aligning the detected landmarks with a mean shape. Finally, we apply the transformation to the image and crop it. The resulting image is then re-sized to a resolution of $96 \times 96$ pixels.

### A.4 NETWORK ARCHITECTURES

Fig. 5 illustrates the network architectures used in our experiments. CN refers to pixel-wise normalization as described in (Karras et al., 2018). FC defines a fully-connected layer. For Leaky ReLU non-linearities, we have used an slope of 0.2. Given that we normalize the images in the range $[-1, 1]$, we use an hyperbolic tangent function as the last layer of the generator. For the discriminator

$d_\gamma(\mathbf{x}, \mathbf{z})$, we use the same architecture showed for $d_\xi(\mathbf{x}, \mathbf{z}, \mathbf{e})$ but removing the input corresponding to $\mathbf{e}$. For the Adam optimizer (Kingma & Ba, 2015) , we used $\alpha = 10^{-4}, \beta_1 = 0.5, \beta_2 = 0.99$ and $\epsilon = 10^{-8}$. Note that the described architectures and hyper-parameters follow standard definitions according to most of GAN/VAEs previous works.

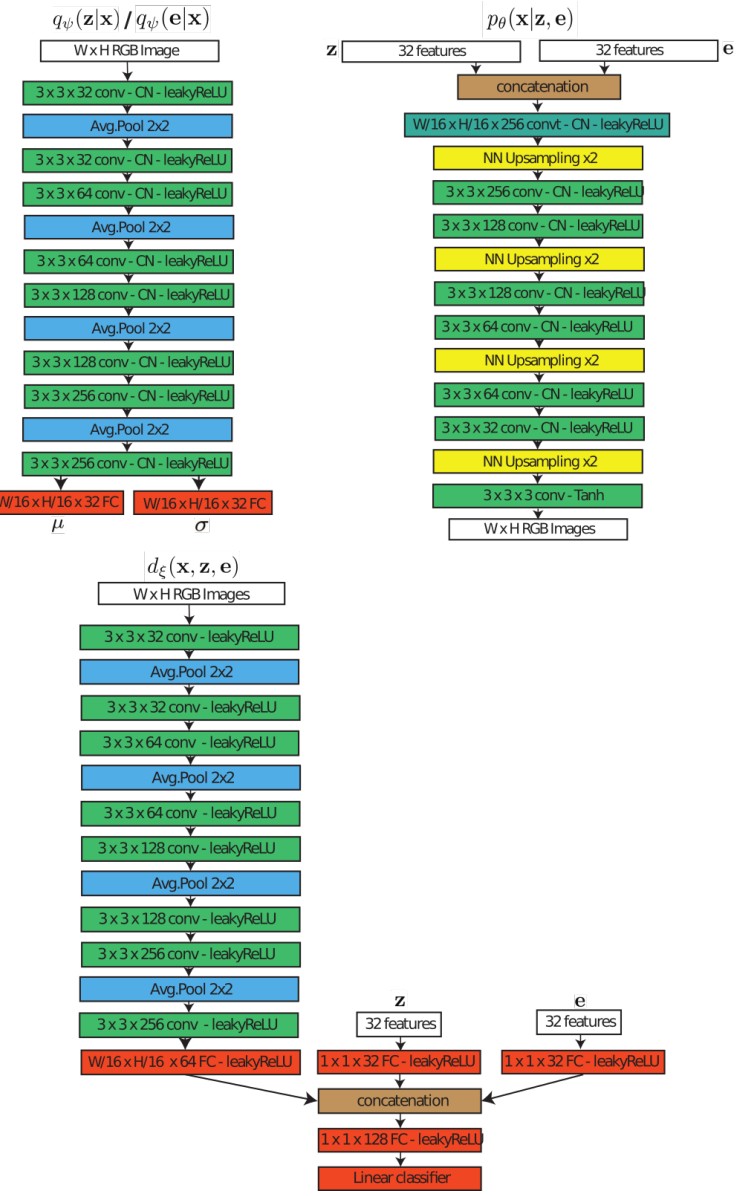

Figure 5: Network architectures used in our experiments

