# OpenReview forum: "Reference-based Variational Autoencoders"
_ICLR.cc/2019/Workshop/LLD — LLD 2019_

### Official Review · AnonReviewer2 · 2019-04-07
**Interesting paper but not well explained**

**Rating:** 3
**Confidence:** 3

**Review:**

The authors provides a training paradigm to leverage "weak supervision" signal via a reference set, which they demonstrate empirically to be effective in regularizing the target factors to encode certain features of interests. This line of work seems quite similar in nature to [1,2,3], who propose to regularize the shared features instead of enforcing them to be a constant.


Pros:
`1. to address the over-regularization problem in traditional VI, the author proposed to use the symmetrized KL loss to encourage the model to make use of the target factor, which is shown to be effective by their experiment.
2. the authors have demonstrated the effectiveness of using a reference set to disentangle target factors from common factors. (see point 2 of Cons)

Cons:
1. the author proposed to use adversarial training to minimize the two KL divergences, but a justification of the necessity of this method is not provided.
2. it is not clear how forcing the target factor of a reference set to be a constant among all data points within the set can help with disentanglement in general. It seems to me to be a heuristic that providing some sort of "weak supervision" will impose some structure in the space of target factor, but I'm not sure to what extent this is effective; e.g. whether certain features will be discarded by "e" and be encoded by "z", or the other way around. Also, have you tried to (1) vary the size of the reference set, e.g. making it smaller, or (2) using multiple reference targets, e.g. with more shared features.

In general, this is an interesting direction. But I'd like to see some intuitive explanation of how this specific regularization help with "disentanglement", which is claimed by the authors.


[1] Unsupervised Learning of Disentangled and Interpretable Representations from Sequential Data
[2] Inferring Identity Factors for Grouped Examples
[3] Disentangled Sequential Autoencoder

---

### Official Review · AnonReviewer1 · 2019-04-08
**Solid paper, with clear discussion and contributions**

**Rating:** 4
**Confidence:** 3

**Review:**

This paper succinctly describes a VAE based method driven by a "reference set" for factor discovery.

I particularly appreciate the detailed experimental section, and extremely thorough comparisons with other architectures. The appendix is also a nice addition, though even the four page version stands on its own merit.

The work itself is quite clear, and the results speak for themselves. They also raise a number of follow-on questions, maybe the authors have explored these in other experiments - none of these are required to be answered, but the author's insights may be useful for other readers of the work (as well as myself).

The selection of the reference set seems important - is it critical that the the factors of interest be constant across every example? How much "noise" in this selection is tolerable? Does choosing different subsets which both hold the variable constant, result in different factors discovered? How does the size of the reference set impact the result generally, is there some threshold below which the method performs far worse?
Are there any suggestions for (perhaps) automated discovery of these constant subsets, or ways to bootstrap this labeling via weak learners and pruning?

The primary things that would improve this paper's rating for me are larger scale experiments, or perhaps non-image data. The model, its baselines, the datasets used, and the experiments completed are all thoroughly spelled out in this paper. Will the authors also release code at some point?

One other paper of interest may be GLSR-VAE (https://arxiv.org/abs/1707.04588) - though not directly comparable to this work, the use of a simple "reward" / label to drive factorization of the latent space seems similar in some ways to the requirement of the reference set to contain the factor of interest.

---

### Decision · Program_Chairs · 2019-04-08
**Acceptance Decision**

Accept